# Bilaterally Symmetrical: To Be or Not to Be?

**Michael C. Corballis**

School of Psychology, University of Auckland, Auckland 1010, New Zealand; m.corballis@auckland.ac.nz

**Abstract:** We belong to a clade of species known as the bilateria, with a body plan that is essentially symmetrical with respect to left and right, an adaptation to the indifference of the natural world to mirror-reflection. Limbs and sense organs are in bilaterally symmetrical pairs, dictating a high degree of symmetry in the brain itself. Bilateral symmetry can be maladaptive, though, especially in the human world where it is important to distinguish between left and right sides, and between left-right mirror images, as in reading directional scripts. The brains of many animals have evolved asymmetries, often but not exclusively in functions not dependent on sensory input or immediate reaction to the environment. Brain asymmetries in humans have led to exaggerate notions of a duality between the sides of the brain. The tradeoff between symmetry and asymmetry results in individual differences in brain asymmetries and handedness, contributing to a diversity of aptitude and divisions of labor. Asymmetries may have their origin in fundamental molecular asymmetries going far back in biological evolution.

**Keywords:** animal asymmetries; bilateral symmetry; bilateria; cerebral asymmetry; handedness; mirror-image discrimination; molecular asymmetry

---

## 1. Introduction

We belong to the vast clade of species known as the bilateria, whose body plan is fundamentally symmetrical with respect to left and right. The two sides of the body are to a high degree left-right mirror images, with limbs and sense organs are arranged in mirrored pairs on opposite sides of the body. Bilateral symmetry has even been taken as proof of the existence of God. Isaac Newton, for example, remarked on the perfect symmetry of the body, making an exception only of the bowels, and thought that it proved "the counsel and contrivance of an Author." William Blake may also have had a celestial author in mind in his 1794 poem, The Tyger:

> Tyger Tyger, burning bright,
> In the forests of the night;
> What immortal hand or eye,
> Could frame thy fearful symmetry.

Until well into the nineteenth century, even the brain was considered symmetrical, with the left side mirroring the right. In 1836, a little-known French physician called Marc Dax read a paper at a conference describing evidence that the left side was dominant for language. The paper was widely distributed but was either ignored or dismissed by the French medical establishment, which firmly held to the principle of symmetry, until further evidence for cerebral asymmetry emerged in the 1860s [1]. Dax's son Gustave then arranged to have his father's paper published in an attempt to establish precedence for the discovery [2].

In evolutionary terms, much is adaptive about bilateral symmetry. The movement of animals depends on having paired limbs, be they legs for walking, wings for flying, flippers for swimming. With one small exception to be discussed later, the natural world is without any systematic biases

one way or the other, so that having limbs of equal length and strength ensures linear movement. Having arms of equal length means equal access to either side of the body for picking things up, plucking fruit, or carrying things—or indeed fighting. The pairing of limbs may well have dictated the symmetrical pairing of sense organs—eyes, ears, nostrils, and of course the sensations arising from the limbs themselves. In a world without left-right bias, animals need to be equally sensitive to events on either side of the body, be they threats or opportunities. As Martin Gardner [3] put it in his book The Ambidextrous Universe:

> The slightest loss of bilateral symmetry, such as the loss of a right eye, would have immediate negative value for the survival of any animal. An enemy could sneak up unobserved on the right! (p. 70)

The symmetry of the limbs and sensory systems probably drove the symmetry of the brain itself, at least insofar as much of brain activity has to do with interpreting the sensory world and organizing appropriate actions.

A perfectly symmetrical brain would also treat left and right as equivalent, and mirror-image patterns as though they were the same. A perfectly symmetrical person, for example, would not be able to tell which hand is which. This can be shown by imagining such a person being asked to hold out the right hand, and doing so correctly. Viewed in a mirror, the person is exactly the same, because mirror reflection leave symmetrical objects unaltered, but is now seen holding out the left hand to the same instruction. Through reduction ad absurdum, it is therefore impossible for the person to comply in consistent fashion. Our symmetrical person would not be able distinguish mirror-image patterns by giving them different labels, so a b could not be labelled a "bee" and a d a "dee" [4]. The symmetrical tiger would correctly regard a prey as the same whether in left or right profile.

Evidence suggests, moreover, that the brain tends to retain its symmetry in spite of asymmetrical experience. A likely mechanism for this is homotopic transfer of learning between the two sides of the brain, so that learned patterns established on one side of the brain are reversed in transfer to the other side [4,5]. This provides for what we termed mirror-image generalization. This is adaptive in the natural world. If a tiger attacks from one side and you survive, you may then be prepared for another attack from either side. Faces and bodies can appear in either profile, which are mirror images, so that if you encounter a person or an animal in one profile it is then generalized to the other.

In the human world, though, this can sometimes be maladaptive. Children learning to read and write often suffer left-right confusions, treating mirror-image letters, such as b and d, or even words like was and saw, as though they were the same [6]. The confusion can persist in spite of asymmetrical training. One of my sons, aged about five, returned from school having proudly printed the entire alphabet and digits from right to left, each one rendered in mirror-image form, despite the fact that he had only been shown them printed normally. (He is now a novelist). The American author Eileen Simpson, who suffered from dyslexia, wrote of her persistent tendency to read the word "was" as "saw" causing her exasperated aunt to exclaim "No. How can you be so stupid? The word is 'was' WASWASWAS" [7].

The problem of telling left from right is not restricted to young children or people with reading problems. According to one survey, 26.2 percent of college students reported some left-right confusion [8,9], while for college professors, perhaps slightly less willing to admit frailty, the proportion was 19.5 percent [10]. In both studies, women reported a higher incidence than did men, but were perhaps simply more honest.

## 2. The Breaking of Symmetry

In spite of the firm belief in the symmetry of the brain through most of the human history, there had long been evidence that the human brain is actually asymmetrical. We are nearly all right-handed, an asymmetry woven into many cultural practices [11] and often a source of prejudice against the 12 percent or so of left-handers. Fossil evidence suggests that even the Neanderthals were right-handed,

perhaps in the same proportion as in humans [12]. Historically, this ubiquitous asymmetry seems to have been regarded as behavioral, or habitual, rather than biological. As Plato put it, "It is due to the folly of nurses and mothers that we have all become limping, so to say, in the hands." The English propagandist Charles Reade [13] was more severe:

> Six thousand years of lop-armed, lop-legged savages, some barbarous, some civilized, have not created a single lop-legged, lop-armed child, and never will. Every child is even and either handed until some grown fool interferes and mutilates it. (p. 175)

As a corrective, John Jackson, a grammar school educator from Belfast, co-founded the British Ambidextral Culture Society in 1903, with the aim of restoring the "natural" symmetry of the hands, as later described in his book Ambidexterity or Two-Handedness and Two-Braineness: An Argument for Natural Development and Rational Education [14]. Lord Baden-Powell, who founded the Boy Scout Movement, supported the Society insisted that scouts shake hands with the left hand. In spite of all this, the majority of people, then as now, are resolutely right-handed, a disposition evident even in the fetus [15]. Handedness is at least partly under genetic control [16], and is surely a function of the brain rather than of the hands themselves.

What seems to have eventually convinced the 19th-century medical establishment that the brain is after all asymmetrical, and that Marc Dax was right, was not the ubiquitous preference for the right hand, but rather further demonstrations that both the production and comprehension of speech are impaired following left-brain damage [17,18]. This led to a remarkable change of emphasis. Gone was the principle of symmetry; the two sides of the brain were now considered so different as to have opposite or complementary functions. This began with the suggestion that perception and emotion might be housed in the right as a counter to language and action on the left, but this morphed into more extreme notions of hemispheric duality. The left hemisphere harnessed humanity, volition, masculinity, and reason, while animality, instinct, femininity, and madness were closeted in the right. The left hemisphere was considered the dominant one, epitomized by the white European male. These developments, and the late 19th-century obsession with brain duality, are documented by the historian Anne Harrington [19], but were largely forgotten shortly after the turn of the 20th century. They were revived in somewhat different guises following the split-brain studies of the 1960s.

These studies were based on patients who had undergone section of corpus callosum and in some cases other forebrain commissures, for the relief of intractable epilepsy. The operation was largely successful in reducing seizures, but effectively disconnected the two sides of the brain, at least with respect to cognitive function. This enabled researchers to test the mental capacities of each side of the brain more or less independently of activity in the other side. The results quickly confirmed that the left side of the brain in these patients was indeed dominant for speech, while in most cases the right side of the brain was essentially mute.

Research also revealed complementary capacities in the disconnected right hemisphere. In his Nobel-Prize address, Roger W. Sperry [20] summarized:

> The right-hemisphere specialties were all, of course, nonverbal, nonmathematical, and nonsequential. They were largely spatial and imagistic, the kind in which a single picture or mental image is worth a thousand words. Examples include reading faces, fitting designs into larger matrices, judging whole circle size from a small arc, discriminating and recalling nondescript shapes, making mental spatial transformations, discriminating musical chords, sorting block sizes and shapes into categories, perceiving wholes from a collection of parts, and the intuitive perception and apprehension of geometric principles. (p. 1224)

The idea that the two sides of the brain had different functions quickly blossomed into a more dramatic duality, although one that was rather different from the 19th-century version. The left side was seen to stand for logic, reason, and propositional thought, the right for intuition, emotion, and appositional thought [21]. The duality quickly gained popularity (e.g., Reference [22]), probably

fanned by the various protest movements of the 1960s—feminism, opposition to the Vietnam war, anti-racism [22]. The left hemisphere was associated with the militant West and cold logic, and the right hemisphere with the supposedly peaceful and creative East. In the popular slogan "make love not war," the right brain evoked love and the left brain war. Betty Edwards' 1979 book Drawing on the Right Side of the Brain [23], along with many other authors, urged release from the tyranny of the left brain in education and even in business, so that the creativity of the right brain could find expression. The idea of the dual brain has persisted well into the 21st century. In his 2009 book The Master and his Emissary, the Scottish psychiatrist Iain McGilchrist [24] characterizes the history of western civilization in terms of alternating left- and right-hemispheric dominance. The right brain was deemed the master and the left brain the emissary, again in opposition to the earlier idea that the left brain is dominant.

We now know that the idea of the dual brain is an exaggeration, as much a product of cultural dichotomies as of neurological fact. For example, right-handedness was long thought to be an aspect of the left-brain's general dominance, so that left-handers were right-brained dominant for language. In fact, the correlation between handedness and language asymmetry is quite low, and most left-handers are in fact left-cerebrally dominant for language. One brain-imaging study showed "typical" left-hemisphere dominance for speech in 88 percent of right-handers and as many as 78 percent of left-handers [25].

Handedness itself is correlated with functional asymmetries in the human brain [26–28] but large scale-studies show no correlation of handedness with structural asymmetries of either grey matter [29] or white matter [30], assessed over widely distributed brain regions. Curiously, though, hand preference is correlated with structural asymmetries in several nonhuman primates, including squirrel monkeys [31], capuchin monkeys [32], and chimpanzees [33].

Handedness aside, even within the brain there appear to be multiple asymmetrical circuits rather than a simple dichotomy). These circuits are uncorrelated, implying different asymmetrical influences. Multiple asymmetries are also evident anatomically. From a sample of 446 individuals, Van Essen et al. [34] parceled the brain into 180 distinct areas and in 128 of these the area on one or other side was, on average, significantly larger than the other. About as many were larger on the left as were larger on the right. To speak of individuals as "left-brained" or "right-brained," still common in popular discourse, makes little sense.

## 3. Differences between Species

Until recently, it was widely held that brain asymmetry was uniquely human, perhaps even defining our species [35–38]—an idea perhaps driven by the fact that language itself is unique to humans. My own view on this has changed, with the welter of evidence for cerebral and behavioral asymmetries that emerged, especially from the early 1990s (see [39], for a review). Some species even show consistent handedness. Around 65 to 70 percent of great apes favor the right hand in various tasks [40,41]), although the incidence is lower than that in humans, which stands at around 90 percent. In some species of parrot, though, about 90 percent prefer the left paw when picking up pieces of food [42]. Marsupials are also predominantly left-handed when feeding, with the incidence rising to 90 percent among those that are bipedal, including kangaroos [43]. This suggests that preference for one or other forelimb may well have emerged with bipedalism, where the forelimbs are no longer involved in bilateral locomotion.

The left-hemispheric specialization for speech and language may well derive from left-hemispheric control of vocalization. This has been demonstrated even in the frog, suggesting an ancestry going back to the very origins of the vocal cords some 170 million years ago [44]. A left-hemispheric advantage for the perception of species-specific vocalizations also occurs in mice [45], cats [46], dogs ([47], domestic horses [48], Californian sea lions [49], rhesus monkeys [50], and Japanese macaques [51]. In chimpanzees, the left temporal planum is larger on the left than on the right [52], an asymmetry well documented in humans (e.g., [53]). This too may reflect an asymmetry in the perception, and perhaps

understanding, of species-specific vocal communication, and may be precursors to asymmetries in language representation.

A right-hemisphere bias has been documented for social responses in a diverse range of vertebrates, for example, in fish, chicks, sheep and monkeys (for review, see [54]). The right hemisphere also seems to mediate social understanding in humans (e.g., [55]). But there is also a negative side, as right hemisphere is the more specialized for aggressive behavior in several species, including anurans [56], lizards [57], chicks [58], baboons [59], and humans [60]. MacNeilage, Rogers and Vallortigara [61] suggest that brain asymmetry was already present when the vertebrates emerged some 500 million years ago, with the left hemisphere oriented more toward action, including feeding and aggression, and the right toward emotion and detection of predators.

Although these and other example show evidence for cerebral asymmetry in many different species, the extent of asymmetry may be larger in humans than in other species, perhaps simply as a consequence of a greater diversity of function. In their study of anatomical asymmetries in humans, Van Essen et al. [34] also examined rodents and primates and found that the number of "parcels" increased from mouse to marmoset to macaque to chimpanzee to human, and so did the proportion of asymmetrical parcels.

## 4. Why Asymmetry?

A number of authors have pointed out that bilateral symmetry can confer disadvantage as well as advantage. Ghirlanda, Frasnelli, and Vallortigara [62], for example, point out that asymmetry can increase efficiency of processing by reducing duplication of brain circuitry and interference between different functions. Asymmetry also creates better use of brain space by providing for increased specialization. In sheer computational terms, symmetry is unnecessarily restrictive: It would be foolhardy at best to try to design a computer while retaining bilateral symmetry among its elements. The brain is a double organ with most of its computational capacity within its two hemispheres rather than straddling them, and it would be wasteful of neural space to duplicate processing in the two. Much of the brain, especially in humans, has to do with internal thinking that has no direct contact with the immediate environment; we think about past events, imagine future ones, draw on more abstract knowledge, and invent different scenarios. The ability to travel mentally in space and time is probably common to many animals [63], and we humans have also evolved language, whose primary function seems to be to communicate about what is in our minds rather than what is present in the immediate environment [64]. This is the property of language known as displacement [65].

The pressure for more neural space and larger brains increased with the demand for more complex processing. The size of the head itself is nevertheless restricted, especially in bipedal animals where the size of the birth canal is restricted by the mechanical demands of upright walking. In humans, this creates what has been termed the "obstetrical dilemma," a hypothesis to explain why childbirth is so difficult, leading to dangerously early birth normally requiring assistance [66]. Nevertheless we need large brains to cope with our complex existence on the planet. The pressure for larger brains in a constrained skull explains why the human brain is exceptionally wrinkled and folded. It may also help explain the multiple asymmetries in the human brain, making better use of the restricted brain space. This is not the only source of brain asymmetry, though, because lateralization is evident even in the brains of small insects, where space is not at a premium,

In spite of the pressures toward asymmetry, the brain seems to retain a moderately high degree of symmetry, at least early in development, which can allow one hemisphere to take over the acquisition of functions if the normally dominant one is damaged. Functions that start out as bilateral can become lateralized during development. A case in point is the fusiform gyrus, which extends from the occipital into the inferior temporal lobe, seems to be host to representations of different categories of visual input, such as faces, objects, words, and scenes. In the chimpanzee and in young children, both sides are involved in face recognition. As children learn to read, though, the left fusiform establishes the visual word form area (VWFA), specialized for the reading of words, while the right hemisphere retains

its capacity for recognizing faces, in the fusiform face area (FFA) opposite the VWFA [67]. The VWFA, because it supplies an asymmetry, may help children overcome mirror-images confusions, which can hinder early reading [5].

The advantages of asymmetry does not easily explain why many species, including humans, show consistency in the direction of asymmetry, while at the same time maintaining a small proportion of individuals with the opposite asymmetry [62]. The relative proportions depend on interactions between predator and prey. For example, if a predator attacks a large group of animals, a small group may escape by fleeing in a direction opposite to the herd, but that advantage only hold so long as that group is a minority. The predator will tend to pursue the larger group to increase the chance of capture, but chances of capture are also relatively small in that group by virtue of its size. Ghirlanda et al. [62] elegantly work out conditions for an evolutionary stable strategy. In humans, left-handers may hold an advantage in sports like fencing or tennis, but only so long as they are a minority.

Frasnelli and Vallortigara [68] note that the proportions showing the same direction of asymmetry ranges from 60 to 90 percent, depending on the task and species, and suggest that common asymmetry may also be driven by the demands of coordination with other asymmetrical individuals, giving examples from aggressive and mating displays among insects. They suggest that the pressure for asymmetry need not be dependent on group behavior, but can apply to inter-individual interactions in solitary species.

## 5. Where did Asymmetry Come from?

The seeming preponderance of bilateral symmetry among animals raises the question of how asymmetry was introduced. In fact, though, asymmetry may be the ancestral condition, predating the bilateria. At the molecular level, biological organisms are fundamentally asymmetrical, suggesting that asymmetry actually came first in biological evolution. DNA is the molecule containing the genetic information governing our growth, and is itself famously asymmetrical, a double helix [69].

One question is whether the asymmetries of living molecules somehow derives from some fundamental asymmetry in the forces of nature. I noted earlier that the natural world is without systematic left-right bias, so that natural events viewed in a mirror seem normal. This is true of three of the fundamental forces of nature. These are the gravitational force that keeps celestial bodies from flying apart, the strong force that holds the nucleus of the atom together, and the electromagnetic force, which is the force between electrically charged particles. These are unaltered by mirror reflection, obeying what physicists call the conservation of parity. There is, however, a fourth fundamental force known as the weak force, responsible for radioactive decay and nuclear fission, in which parity is not conserved. This was discovered in 1956 to have a fundamental asymmetry, so that when viewed in a mirror it does not reveal its true nature [70].

An early conjecture, known as the Vester-Ulbrech hypothesis, was that asymmetries in beta radiation bombarding molecules that were precursors of life would gradually destroy one of the two mirror image forms (enantiomers), leaving the other to survive [71]. Attempts to test this by bombarding organic material with beta rays had been largely negative [72]. In 2014, though, two physicists did find that bombardment of bromocamphor molecules with spin-polarized electrons caused left-handed molecules to disintegrate slightly more often than right-handed ones [73].

The difference was tiny and took a long time to emerge. One of the authors of the study, Joan Dreiling, was quoted as saying, "The scale of the asymmetry is as though we flip 20,000 coins again and again, and on average, 10,003 of them land on heads while 9997 land on tails." Could we really owe our asymmetrical molecules, and even our existence on the planet, to such a small, cosmic asymmetry? It suggests an extra challenge to Albert Einstein's famous remark that, "God does not play dice with the universe." God may not only play dice, but may take note of how the dice fall.

In terms of biological evolution, then, symmetry seems not to be the default condition, but rather one constructed from an asymmetrical molecular base. As an example, Vopalensky et al. [74] give an account of how embryonic development in a marine annelid (Platynereis dumerilii) progresses

during cell division from a spiral form to a bilaterally symmetrical one. The actual mechanism of this transition seems to be unknown.

## 6. The Evolutionary Tradeoff

From the beginning of life itself, there has been a tradeoff between symmetry and asymmetry. Perhaps because of some asymmetrical cosmic radiation, in which parity was not conserved, biological molecules gained asymmetric structures. Aside from the asymmetry of the weak force, though, the forces of nature preserve parity, and we might as well have inhabited the looking glass world as the actual one. Symmetry was then forged from asymmetry, resulting in the emergence of the bilateria and the predominance of bilaterally symmetrical species. Practical considerations then led to the introduction of asymmetrical features.

The brain itself is both symmetrical and asymmetrical, with an uneasy balance between the two. Bilateral symmetry is adaptive in perceiving and acting in a natural world without systematic left-right bias, at least with respect normal experience. Symmetry, though, seems readily abandoned where asymmetry is more adaptive. This may apply especially in animals requiring behaviors beyond merely reacting to environmental impact, including operations that are manipulative rather than reactive. In internally-generated operations, bilateral symmetry could be unwieldy or restrictive, creating either duplication of circuits in the two hemispheres, or unitary circuits that must somehow retain symmetry. A useful analogy is vehicles of transport. Simple horse-drawn carts can be symmetrical to ensure optimal motion and least resistance, but when motors are introduced symmetry is no longer retained internally. To create perfectly symmetrical engines would be an unnecessary challenge.

Symmetry is more readily abandoned in the internal organs than in other parts of the body, including the brain and limbs. As the German physicist and philosopher Hermann Weyl wrote in his 1952 book Symmetry [75]:

> Factors in phylogenetic evolution that tend to introduce heritable differences between left and right are likely to be held in check by the advantages an animal derives from the bilateral formation of its organs of motion, cilia, muscles and limbs: in case of their asymmetrical development a screw-wise instead of a straightforward motion would naturally result. This may help to explain why our limbs obey the laws of symmetry more than our inner organs. (p. 27)

Asymmetry in the internal organs of the body makes for more efficient packaging and probably more efficient function if symmetry is abandoned. As Newton saw it, bilateral symmetry holds "with the exception only of the bowels," but it is not just the bowels; most of our vital organs are arranged asymmetrically, such as the leftward placement of the heart or and stomach, or the rightward displacement of the gall bladder and liver. There is still a hint of symmetry in the occurrence of situs inversus totalis, in which the internal organs are left-right reversed. This is very infrequent, affecting about one individual in 10,000 [76]. When it does occur, it seems to come about from a rare loss of directional information, so that the direction of situs becomes a matter of chance [77,78].

The uneasy balance between symmetry and asymmetry in the human brain is betrayed by individual variation. Unlike situs inversus, reversals of handedness and cerebral asymmetry are relatively common, somewhere around 12-13 percent, although again there is some reason to suppose that left-handedness, at least, is a result of the loss of directional information rather than its reversal. That is, around 25 percent of the population lack the biological disposition to be right-handed, so they are divided equally into left- and right-handers, with a small proportion perhaps better classified as ambidextrous. If right-handedness depends on a gene with one allele coding for right-coding for its absence, then balance between the two could be maintained by a heterozygotic advantage, favoring those with one of each allele over homozygotes with two identical alleles [79]. The gene is assumed dominant, so that heterozygotes are right-handed. The heterozygotic advantage may be nature's way of achieving the compromise between symmetry and asymmetry, and mhandedness and another

aintaining constant proportions of left- and right-handers in the population [80]. It may also be the mechanism underlying the uneven balance between the different directions of asymmetry [62].

The idea of a heterozygotic advantage provides for stability in the distribution of laterality, with optimal proportions of 50 percent for heterogyotics, and 25 percent for each homozygote, with 12.5 percent of those lacking the lateralizing allele showing reversal (i.e., left-handers, or those with reverse brain asymmetries). Indeed this proportion of left-handedness seems to be roughly constant over all peoples over the centuries, although it may also be influenced by culture. For example, the percentage of children in Australia and New Zealand writing with the left hand rose from about 2 percent at the turn of the 19th century to about 13 percent in the 1960s [81], and a similar increase occurred in the United States between the 1920s and the 1960s [82]. Natural variation in asymmetry provides for individual differences, offering what Szathmàry [83] called the "negotiated division of labor." For example those heterozygotes with double lateralizing alleles may be especially disposed to verbal capability but perhaps deficient in navigation, while those lacking the lateralizing allele may have superior spatial skills but perhaps a disposition to language problems—or even dyslexia.

This theory may be wrong in detail, but perhaps right in spirit. It transpires that there are probably several genes, not just one, underling handedness itself [84], and brain asymmetry more generally [30], although each may operate in similar fashion. As a general principle, it may be that genetic influences on asymmetry control whether an asymmetry is present or absent, but not the direction of the asymmetry (e.g., [85]), and in that respect at least brain asymmetry may resemble the asymmetry of the internal organs. Yet these asymmetries appear to be independent. In one study, 15 out of 16 individuals with situs inversus were right-handed [86], while another study of showed three people with situs inversus showed all to be left-cerebrally dominant for language, with larger temporal plana on the left, as in the majority with normal situs [87]. To return to Newton, it is not just the bowels that are asymmetrical, but also the brain, albeit on a different trajectory.

Mirror images loom large in culture and biology, from isomeric molecules to left- and right-handedness to left- and right-brains. They are imbued with cultural and sometimes cosmic significance. The important dichotomy, though, may be not so much between left and right as between symmetry and asymmetry itself. The conflict between symmetry and asymmetry plays out not only in the brain but in its external manifestations, such as art and architecture. On that note, I leave the last word to Madame de Maintenon, second wife of Louis XIV of France, who wrote of her husband that "he thinks of nothing but grandeur, magnificence, and symmetry." But symmetry meant that windows and doors in the palace were placed opposite one another, creating draughts and affecting her health. She went on famously to declare, "you must perish in symmetry" ([88]).

**Funding:** This research received no external funding.

**Conflicts of Interest:** The authors declares no conflict of interest.

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
