# Peer review of "Bilaterally Symmetrical: To Be or Not to Be?"

_symmetry, doi:10.3390/sym12030326_

Round 1

Reviewer 1 Report

Lines 13-14: The cumulative evidence from numerous recent studies clearly showed that the brains of many animals have evolved asymmetries in functions DEPENDENT on sensory input or immediate reaction to the environment (for a review see e.g., MacNeilage, P.F., Rogers, L.J., Vallortigara, G., 2009. Origins of the left and right brain. Sci.Am. 301, 60–67; Rogers, L.J., 2017a. Considering side biases in vigilance and fear. Animal Sentience 15, 4; Rogers, L.J., 2017b. Eye and ear preferences. In: Rogers, L.J., Vallortigara, G. (Eds.), Lateralized Brain Functions—Methods in Human and Non-Human Species. Springer, New York, pp. 79–102.).

Lines 127-131: It is not clear to me which reference this information belongs to.

Lines 178-180: For a balanced, unbiased discussion the opposite results should be mentioned here too. Handedness has been shown to correlate with the enlargement of the cortical representation of the hand in the contralateral motor cortex (Nudo, R. J., Jenkins, W. M., Merzenich, M. M., Prejean, T. & Grenda, R. 1992 Neurophysiological correlates of hand preference in primary motor cortex of adult squirrel monkeys. J. Neurosci. 12, 2918–2947). In non-human primates, a correlation between structural asymmetry of the precentral gyrus and hand preference has been found (Hopkins, W. D. & Cantalupo, C. 2004 Handedness in chimpanzees (Pan troglodytes) is associated with asymmetries of the primary motor cortex but not with homologous language areas. Behav. Neurosci. 118, 1176–1183; Dadda, M., Cantalupo, C. & Hopkins, W. D. 2006 Further evidence of an association between handedness and neuroanatomical asymmetries in the primary motor cortex of chimpanzees (Pan troglodytes). Neuropsychologia 44, 2582–2586; Phillips, K. & Sherwood, C. S. 2005 Primary motor cortex asymmetry correlates with handedness in capuchin monkeys (Cebus apella). Behav. Neurosci. 119, 1701–1704)

Lines 208-210: More species are showing a left-hemispheric advantage for the perception of species-specific vocalizations, e.g., cats (Siniscalchi, M., Laddago, S., & Quaranta, A. (2016). Auditory lateralization of conspecific and heterospecific vocalizations in cats. Laterality: Asymmetries of Body, Brain and Cognition, 21(3), 215-227), dogs (Siniscalchi, M., Quaranta, A., & Rogers, L. J. (2008). Hemispheric specialization in dogs for processing different acoustic stimuli. PLoS One, 3(10)), horses (Basile, M., Boivin, S., Boutin, A., Blois-Heulin, C., Hausberger, M., & Lemasson, A. (2009). Socially dependent auditory laterality in domestic horses (Equus caballus). Animal cognition, 12(4), 611-619), and sea lions (Böye, M., Güntürkün, O., & Vauclair, J. (2005). Right ear advantage for conspecific calls in adults and subadults, but not infants, California sea lions (Zalophus californianus): hemispheric specialization for communication?. European Journal of Neuroscience, 21(6), 1727-1732.).

Lines 215-216: Again, not only fish, chicks, sheep and monkeys demonstrated a right-hemisphere bias has been documented for social responses. This bias has been found in numerous species (reviewed in many papers, e.g., Forrester, G.S., Todd B.K., 2018. A comparative perspective on lateral biases and social behaviour. In: Forrester, G.S., Hopkins, W.D., Hudry, K., Lindell, A. (Eds.), Progress in Brain Research, vol. 238. Cerebral Lateralization and Cognition: Evolutionary and Developmental Investigations of Behavioral Biases. Elsevier, pp 377–403; Rosa Salva, O., Regolin, L., Mascalzoni, E., Vallortigara, G., 2012. Cerebral and behavioural asymmetries in animal social recognition. Comp. Cogn. Behav. Rev. 7, 110–138.). Please, add more species or rephrase, for example, like this - “A right-hemisphere bias has been documented for social responses in a diverse range of vertebrates, for example, in fish, chicks, sheep and monkeys”.

General comments

I found this to be a well-written manuscript introducing a reader to an impressive theoretical investigation.  The study is devoted to structural and functional asymmetry in organisms in general, and brain lateralization in vertebrates in particular. A theoretical analysis presented in the manuscript is a result of an interdisciplinary approach based on the evidence from history, physics and biology. The author discusses important issues such as an origin of asymmetry in biological evolution, advantages and disadvantages of asymmetry, and the fundamental significance of dichotomy between symmetry and asymmetry.

A plethora of empirical evidence is discussed to generate new hypotheses. However many important studies were missed out in the course of reasoning. The existing theories about the emergence of brain lateralization simply have to be mentioned (e.g., Ghirlanda, S., Frasnelli, E., Vallortigara, G., 2009. Intraspecific competition and coordination in the evolution of lateralization. Philos. Trans. R. Soc. Lond. B Biol. Sci. 364, 861–866; MacNeilage, P.F., Rogers, L.J., Vallortigara, G., 2009. Origins of the left and right brain. Sci.Am. 301, 60–67; Frasnelli, E., & Vallortigara, G. (2018). Individual-level and population-level lateralization: Two sides of the same coin. Symmetry, 10(12), 739.). There is no need to discuss them in-depth but some mentioning is required to make the manuscript more balanced.  Please see also my specific comments on the missing references.

Author Response

Thank you for all those comments. I have altered the Abstract to remove the suggestion that asymmetries do not arise in sensory systems, and tried to make it clear that the information of 19th-century ideas about the dual brain are referenced in the book by Harrington.

I incorporated the references you gave me on cortical correlates of handedness, and also the references on the many examples of asymmetries for vocalization in nonhuman species. I added the references you gave me on right-hemispheric asymmetries for social responses, and for good measure added specific references on left-hemispheric asymmetries for aggressive behavior. Thank you so much--I had clearly been negligent on document asymmetries in nonhuman species, and I think it is now much more complete.

I also introduced the three articles you mention on theories about the emergence of asymmetry, which involved some reorganization of the section on "Why asymmetry?" This also allowed me to address the question of why a minority of individuals reverse the asymmetry shown by the majority.

I am very much indebted to you for the detailed review, and the multitude of references you supplied!

I hope my responses are not too cursory--I was given 5 days to revise and resubmit! I think the manuscript itself deals with the changes quite extensively.

Reviewer 2 Report

This paper begins with an interesting and very readable summary of the history ideas and research on brain asymmetry in humans. It them moves on to discuss some examples of lateralization in non-human species - in this section it would be preferable to cite some of the original research instead of simply self-citing a review paper – and arrives at the conclusion that the difference between asymmetry in humans and other species is “one of degree, not of kind”. See my point below – exactly what is meant by a difference in degree needs to be explained clearly.

Although this paper does not say anything particularly new, it is a well-written summary of the author’s position on a topic on which he is an expert.

13-14: Here it is said that brain asymmetries have evolved especially in functions not dependent on sensory input but does the evidence does support this? I don’t think this point is discussed sufficiently in the main text of the paper.

219: Here some references, or at least a summary review, on the right hemisphere’s specialisation for aggression should be cited.

228: By saying the difference in asymmetry between humans and non-human species is “one of degree, not of kind” do you mean that more functions are lateralized in humans or that the laterality of particular functions is stronger in humans? I disagree if you mean the latter, since asymmetry of a number of functions (in chicks, for example) is just as strong as found in humans. Anyway, please make it clear exactly what you mean by a difference in “degree”.

251-253: The same rule applies to brains of various sizes in different species. Even the small brains of insects are lateralized, and so it may be said that even they make the most of what they have. In other words, I don’t think the pressure to reduce redundancy and duplication applies any more to humans than it does to other species.

251: Correct explain” to “explains”.

276: Correct “forced” to “forces”.

297: Correct “place” to “play”.

298: Replace “they” by “the dice”.

317: But many species have been shown to react differently to stimuli seen on their left versus right side. Of course, this may not always mean that perception per se differs for inputs from the left and right eyes but there is evidence that the left and right eyes of starlings differ in the number of particular types of photoreceptor cells (Hart et al. Curr. Biol. 2000, 10, 115-117). In other words, I think there is still much research needed before we can say that visual asymmetries result from symmetrical, or asymmetrical, perception.

339: The stomach is usually placed on the left side of the body. The liver is on the right side.

363: Cite a reference to support the statement that handedness is influenced by culture.

372: Also cite a reference here to research showing genetic influences on the presence or absence of asymmetry and not on its direction.

Author Response

I removed the comment about "degree but not in kind" and also the suggestion that asymmetries do not arise in sensory systems" (a point also raised by reviewer 1).

You mention references to right-hemisphere specialization for aggression, but don't provide them--I added some anyway!

I qualified the suggestion that asymmetry may be due to the restriction on neural space.

I fixed the typos and relocated the stomach on the left, and mentioned the biases of the liver and gall bladder to the right. Thank you!

I took the liberty of not referring to asymmetries in photocells in starlings, because I think it takes us beyond the themes of the article, and in any case still needs "much research."

I added a couple of references on cultural effects on handedness. I also added a reference showing a genetic influence over degree but not direction of handedness. This is still fairly conjectural.

Thank you so much for the helpful comments!  I hope my replies are not too cursory but I seem to be under pressure to return the manuscript--I was given 5 days! But  think I have dealt with the comments in the revision itself.